# VTBIS: Vision Transformer for Biomedical Image Segmentation

## Abstract

*In this paper, we propose a novel network named Vision Transformer for Biomedical Image Segmentation (VTBIS). Our network splits the input feature maps into three parts with $1 \times 1$, $3 \times 3$ and $5 \times 5$ convolutions in both encoder and decoder. Concat operator is used to merge the features before being fed to three consecutive transformer blocks with attention mechanism embedded inside it. Skip connections are used to connect encoder and decoder transformer blocks. Similarly, transformer blocks and multi scale architecture is used in decoder before being linearly projected to produce the output segmentation map. We test the performance of our network using Synapse multi-organ segmentation dataset, Automated cardiac diagnosis challenge dataset, Brain tumour MRI segmentation dataset and Spleen CT segmentation dataset. Without bells and whistles, our network outperforms most of the previous state of the art CNN and transformer based models using Dice score and the Hausdorff distance as the evaluation metrics.*

## 1. Introduction

Deep Convolutional Neural Networks has been highly successful in medical image segmentation. U-Net (Ronneberger et al., 2015) based architectures use a symmetric encoder-decoder network with skip-connections. The limitation of CNN-based approach is that it is unable to model long-range relation, due to the regional locality of convolution operations. To tackle this problem, self attention mechanism was proposed (Schlemper et al., 2019) and (Wang et al., 2018). Still, the problem of capturing multi-scale contextual information was not solved which leads not so accurate segmentation of structures with variable shapes and scales (e.g. brain lesions with different sizes).

An alternative technique using Transformers are better suited at modeling global contextual information. Vision Transformer (ViT) (Dosovitskiy et al., 2020) splits the image into patches and models the correlation between these patches as sequences with Transformer, achieving better speed-performance trade-off on image classification than previous state of the art image recognition methods. DeiT (Touvron et al., 2020) proposed a knowledge distillation method for training Vision Transformers.

An extensive study was done by (Bakas et al., 2018) to find the best algorithm for segmenting tumours in brain. Medical images from CT and MRI are in 3 dimensions, thus making volumetric segmentation important. Çiçek et al. (2016) tackled this problem using 3d U-Net. Densely-connected volumetric convnets was used (Yu et al., 2017) to segment cardiovascular images. A comprehensive study to evaluate segmentation performance using Dice score and Jaccard index was done by (Eelbode et al., 2020).

## 2. Related Work

### 2.1. Convolutional Neural Network

Earlier work for medical image segmentation used some variants of the original U-shaped architecture (Ronneberger et al., 2015). Some of these were Res-UNet (Xiao et al., 2018), Dense-UNet (Li et al., 2018) and U-Net++ (Zhou et al., 2018). These architectures are quite successful for various kind of problems in the domain of medical image segmentation.

### 2.2. Attention Mechanism

Self Attention mechanism (Wang et al., 2018) has been used successfully to improve the performance of the network. (Schlemper et al., 2019) used skip connections with additive attention gate in U-shaped architecture to perform medical image segmentation. Attention mechanism was first used in U-Net (Oktay et al., 2018) for medical image segmentation. A multi-scale attention network (Fan et al., 2020) was proposed in the context of biomedical image segmentation.

(Jin et al., 2020) used a hybrid deep attention-aware network to extract liver and tumor in ct scans. Attention module was added to U-Net module to exploit full resolution features for medical image segmentation (Li et al., 2020). A similar work using attention based CNN was done by (Liu et al., 2020) in the context of schemic stroke disease. A multi scale self guided attention network was used to achieve state of the art results (Sinha and Dolz, 2020) for medical image

segmentation.

## 2.3. Transformers

Transformer first proposed by (Vaswani et al., 2017) have achieved state of the art performance on various tasks. Inspired by it, Vision Transformer (Dosovitskiy et al., 2020) was proposed which achieved better speed-accuracy tradeoff for image recognition. To improve this, Swin Tranformer (Liu et al., 2021) was proposed which outperformed previous networks on various vision tasks including image classification, object detection and semantic segmentation.

(Chen et al., 2021), (Valanarasu et al., 2021) and (Hatamizadeh et al., 2021) individually proposed methods to integrate CNN and transformers into a single network for medical image segmentation. Transformer along with CNN are applied in multi-modal brain tumor segmentation (Wang et al., 2021) and 3D medical image segmentation (Xie et al., 2021).

Our main contributions can be summarized as:

• We propose a novel network incorporating attention mechanism in transformer architecture along with multi scale module in the context of medical image segmentation.

• Our network outperforms previous state of the art CNN based as well as transformer based architectures on various datasets.

• We present the ablation study showing our network performance is generalizable hence can be incorporated to tackle other similar problems.

## 3. Method

### 3.1. Dataset

**1. Synapse multi-organ segmentation dataset** - We use 30 abdominal CT scans in the MICCAI 2015 Multi-Atlas Abdomen Labeling Challenge, with 3779 axial contrast-enhanced abdominal clinical CT images in total.

**2. Automated cardiac diagnosis challenge** - The chest CT scan of each patient is manually annotated with ground truth for left ventricle (LV), right ventricle (RV) and myocardium (MYO).

**3. Spleen CT segmentation** - For task 9 of MSD challenge, 20 CT volumes with spleen body annotation are used.

**4. Brain Tumor Segmentation** - 3D MRI dataset used in the experiments is provided by the BraTS 2019 challenge (Menze et al., 2014) and (Bakas et al., 2018).

### 3.2. Network Architecture

Suppose an image is given $x \in R^{H \times W \times C}$ with a spatial resolution of $H \times W$ and $C$ number of channels. The goal is to predict the pixel-wise label of size $H \times W$ for each image. We start by performing tokenization by reshaping the input $x$ into a sequence of flattened 2D patches $x_p^i \in R(i = 1, .., N)$,

where each patch is of size $P \times P$ and $N = (H \times W)/P^2$ is the number of patches present in the image.

We convert the vectorized patches $x_p$ into a latent $D$-dimensional embedding space using a linear projection vector. We use patch embeddings to make sure the positional information is present as shown in Equation 1:

$$\mathbf{z}_0 = \left[ \mathbf{x}_p^1 \mathbf{E}; \mathbf{x}_p^2 \mathbf{E}; \cdots ; \mathbf{x}_p^N \mathbf{E} \right] + \mathbf{E}_{pos} \qquad (1)$$

where $E \in R^{(P^2 C)} \times D$ denotes the patch embedding projection, and $E_{pos} \in R^{N \times D}$ denotes the position embedding.

After the embedding layer, we use multi scale context block followed by a stack of transformer blocks (Dosovitskiy et al., 2020) made up of multiheaded self-attention (MSA) and multilayer perceptron (MLP) layers as shown in Equation 2 and Equation 3 respectively:

$$\mathbf{z}_i' = \text{MSA} \left( \text{Norm} \left( \mathbf{z}_{i-1} \right) \right) + \mathbf{z}_{i-1} \qquad (2)$$

$$\mathbf{z}_i = \text{MLP} \left( \text{Norm} \left( \mathbf{z}_i' \right) \right) + \mathbf{z}_i' \qquad (3)$$

Where Norm represents layer normalization, MLP is made up of two linear layers and $i$ is the individual block. A MSA block is made up of $n$ self-attention (SA) heads in parallel.

The structure of Transformer layer used in this work is illustrated in Figure 1:

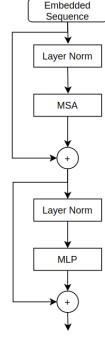

Figure 1. Schematic of the Transformer layer used in this work.

The output sequence of Transformer $z_L \in R^{d \times N}$ is first reshaped to $d \times H/8 \times W/8 \times D/8$ . A convolution block is used to reduce the channel dimension from $d$ to $K$ . This helps in reducing the computational complexity. Upsampling operations and successive convolution blocks are the used to get back a full resolution segmentation result $R \in R^{H \times W \times D}$. Skip-connections are used to fuse the encoder features with the decoder by concatenation to get more contextual information.

In the encoder part, the input image is split into patches and fed into linear embedding layer. The feature map is splitted into N parts along with the channel dimension. The

individual features are fused before being passed to the transformer blocks. The decoder block is comprised of transformer blocks followed by a similar split and concat operator. Linear projection is used on the feature maps to produce the segmentation map. Skip connections are used between the encoder and decoder transformer blocks to provide an alternative path for the gradient to flow thus speeding up the training process.

The detailed architecture of our network as well as the intermediate skip-connections is shown in Figure 2:

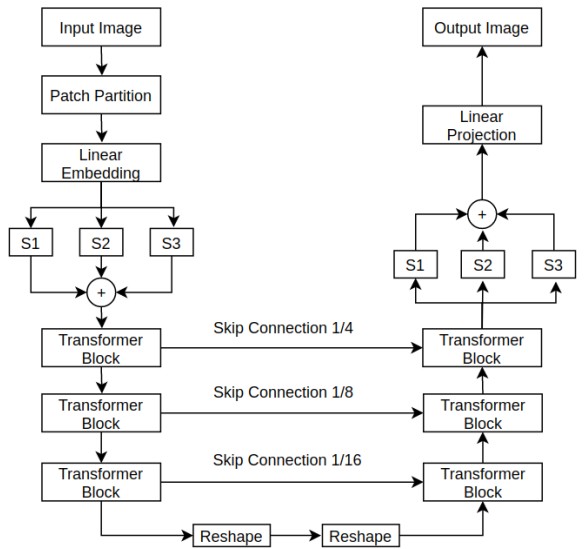

Figure 2. Overview of our model architecture. Output sizes demonstrated for patch dimension N = 16 and embedding size C = 768. We extract sequence representations of different layers in the transformer and merge them with the decoder using skip connections.

Similar to the previous works (Hu et al., 2019), selfattention is computed as defined in Equation 4:

$$\text{MSA}(Q, K, V) = \text{Sof}tMax\left(\frac{QK^T}{\sqrt{d}} + B\right)V \quad (4)$$

where $Q, K, V \in R^{M^2 \times d}$ denote the query, key and value matrices. $M^2$ and $d$ denotes the number of patches in a window and the dimension of the query. The values in $B$ are taken from the random bias matrix denoted by $B \in R^{(2M-1) \times (2M+1)}$

The output of MSA is defined as in Equation 5:

$$\text{TMSA}(\mathbf{z}) = [\text{MSA}_1(z); \text{MSA}_2(z); \dots; \text{MSA}_n(z)]\,\mathbf{W}_{tmsa} \quad (5)$$

Where $W_{tmsa}$ represents the learnable weight matrices of different heads (SA).

## 3.3. Loss Function

Our loss function is a combination of dice and cross entropy terms which is calculated in voxel-wise manner as defined in Equation 6:

$$\mathcal{L} = 1 - \frac{2}{J}\sum_{j=1}^{J}\frac{\sum_{i=1}^{I}G_{i,j}Y_{i,j}}{\sum_{i=1}^{I}G_{i,j}^2 + \sum_{i=1}^{I}Y_{i,j}^2} - \frac{1}{I}\sum_{i=1}^{I}\sum_{j=1}^{J}G_{i,j}\log Y_{i,j} \quad (6)$$

where $I$ is the number of voxels, $J$ is the number of classes, $Y_{i,j}$ and $G_{i,j}$ denote the probability output and one-hot encoded ground truth for voxel $i$ of class $j$.

## 3.4. Evaluation Metrics

The segmentation accuracy is measured by the Dice score and the Hausdorff distance (95%) metrics for enhancing tumor region (ET), regions of the tumor core (TC), and the whole tumor region (WT).

## 3.5. Implementation Details

Our model is trained using Pytorch deep learning framework. The learning rate and weight decay values used are 0.00015 and 0.005, respectively. We use batch size value of 16 and ADAM optimizer to train our model.

We use a random crop of $128 \times 192 \times 192$ and mean normalization to prepare our model input. The input image size and patch size are set as $224 \times 224$ and 4, respectively. As a model input, we use the 3D voxel by cropping the brain region. The following data augmentation techniques are applied:

1. Random cropping of the data from $240 \times 240 \times 155$ to $128 \times 128 \times 128$ voxels;

2. Flipping across the axial, coronal and sagittal planes by a probability of 0.5

3. Random Intensity shift between [-0.05, 0.05] and scale between [0.5, 1.0].

## 4. Results

We report the average DSC and average Hausdorff Distance (HD) on 8 abdominal organs (aorta, gallbladder, spleen, left kidney, right kidney, liver, pancreas, spleen, stomach) with a random split of 20 samples in training set and 10 sample for validation set using Synapse multi-organ CT dataset in Table 1.

We report the average DSC with a random split of 70 training cases, 20 cases for validation and 10 for testing using ACDC dataset in Table 2:

We conduct the five-fold cross-validation evaluation on the BraTS 2019 training set. The quantitative results is presented in Table 3.

Table 1. Comparison on the Synapse multi-organ CT dataset (average dice score %, average hausdorff distance in mm, and dice score % for each organ). The best results are highlighted in bold.

| Encoder | Decoder | DSC | HD | Aorta | GB | Kid(L) | Kid(R) | Liver | Panc |
|---|---|---|---|---|---|---|---|---|---|
| V-Net | V-Net | 68.81 | - | 75.34 | 51.87 | 77.10 | 80.75 | 87.84 | 40.05 |
| DARR | DARR | 69.77 | - | 74.74 | 53.77 | 72.31 | 73.24 | 94.08 | 54.18 |
| R50 | U-Net | 74.68 | 36.87 | 84.18 | 62.84 | 79.19 | 71.29 | 93.35 | 48.23 |
| R50 | AttnUNet | 75.57 | 36.97 | 55.92 | 63.91 | 79.20 | 72.71 | 93.56 | 49.37 |
| ViT | None | 61.50 | 39.61 | 44.38 | 39.59 | 67.46 | 62.94 | 89.21 | 43.14 |
| ViT | CUP | 67.86 | 36.11 | 70.19 | 45.10 | 74.70 | 67.40 | 91.32 | 42.00 |
| R50-ViT | CUP | 71.29 | 32.87 | 73.73 | 55.13 | 75.80 | 72.20 | 91.51 | 45.99 |
| TransUNet | TransUNet | 77.48 | 31.69 | 87.23 | 63.13 | 81.87 | 77.02 | 94.08 | 55.86 |
| SwinUnet | SwinUnet | 79.13 | 21.55 | 85.47 | 66.53 | 83.28 | 79.61 | 94.29 | 56.58 |
| VTBIS | VTBIS | **80.45** | **21.24** | **86.41** | **66.80** | **83.59** | **80.12** | **94.56** | **56.90** |

Table 2. Comparison on the ACDC dataset using DSC evaluation metric(%). The best results are highlighted in bold.

| Framework | Average | RV | Myo | LV |
|---|---|---|---|---|
| R50-U-Net | 87.55 | 87.10 | 80.63 | 94.92 |
| R50-AttnUNet | 86.75 | 87.58 | 79.20 | 93.47 |
| ViT-CUP | 81.45 | 81.46 | 70.71 | 92.18 |
| R50-ViT-CUP | 87.57 | 86.07 | 81.88 | 94.75 |
| TransUNet | 89.71 | 88.86 | 84.53 | 95.73 |
| VTBIS | **90.34** | **89.03** | **85.32** | **95.94** |

Table 3. Comparison on the BraTS 2019 validation set. DS represents Dice score and HD repesents Hausdorff distance. The best results are highlighted in bold.

| Method | ET(DS%) | WT(DS%) | TC(DS%) | ET(HD mm) | WT(HD mm) | TC(HD mm) |
|---|---|---|---|---|---|---|
| 3D U-Net | 70.86 | 87.38 | 72.48 | 5.062 | 9.432 | 8.719 |
| V-Net | 73.89 | 88.73 | 76.56 | 6.131 | 6.256 | 8.705 |
| KiU-Net | 73.21 | 87.60 | 73.92 | 6.323 | 8.942 | 9.893 |
| Attention U-Net | 75.96 | 88.81 | 77.20 | 5.202 | 7.756 | 8.258 |
| Li et al | 77.10 | 88.60 | 81.30 | 6.033 | 6.232 | 7.409 |
| TransBTS w/o TTA | 78.36 | 88.89 | 81.41 | 5.908 | 7.599 | 7.584 |
| TransBTS w/ TTA | 78.93 | 90.00 | 81.94 | **3.736** | **5.644** | 6.049 |
| VTBIS | **79.24** | **90.28** | **82.23** | 3.706 | 5.621 | **7.129** |

We compare the performance of our model against CNN based networks for the task of brain tumour segmentation in Table 4.

Table 4. Cross validation results of brain tumour Segmentation task. DSC1, DSC2 and DSC3 denote average dice scores for the Whole Tumour (WT), Enhancing Tumour (ET) and Tumour Core (TC) across all folds. For each split, average dice score of three classes are used. The best results are highlighted in bold.

| Fold | Split-1 | Split-2 | Split-3 | Split-4 | Split-5 | DSC1 | DSC2 | DSC3 | Avg. |
|---|---|---|---|---|---|---|---|---|---|
| VNet | 64.83 | 67.28 | 65.23 | 65.2 | 66.34 | 75.96 | 54.99 | 66.38 | 65.77 |
| AHNet | 65.78 | 69.31 | 65.16 | 65.05 | 67.84 | 75.8 | 57.58 | 66.50 | 66.63 |
| Att-UNet | 66.39 | 70.18 | 65.39 | 66.11 | 67.29 | 75.29 | 57.11 | 68.81 | 67.07 |
| UNet | 67.20 | 69.11 | 66.84 | 66.95 | 68.16 | 75.03 | 57.87 | 70.06 | 67.65 |
| SegResNet | 69.62 | 71.84 | 67.86 | 68.52 | 70.43 | 76.37 | 59.56 | 73.03 | 69.65 |
| VTBIS | **70.92** | **73.84** | **71.05** | **72.29** | **72.43** | **79.52** | **60.90** | **76.11** | **71.98** |

In Table 5, We compare the performance of our network against previous state of the art for the task of spleen segmentation.

Table 5. Cross validation results of spleen segmentation task. For each split, we provide the average dice score of fore-ground class. The best results are highlighted in bold.

| Fold | Split-1 | Split-2 | Split-3 | Split-4 | Split-5 | Avg. |
|---|---|---|---|---|---|---|
| VNet | 94.78 | 92.08 | 95.54 | 94.73 | 95.03 | 94.43 |
| AHNet | 94.23 | 92.10 | 94.56 | 94.39 | 94.11 | 93.87 |
| Att-UNet | 93.16 | 92.59 | 95.08 | 94.75 | 95.81 | 94.27 |
| UNet | 92.83 | 92.83 | 95.76 | 95.01 | 96.27 | 94.54 |
| SegResNet | 95.66 | 92.00 | 95.79 | 94.19 | 95.53 | 94.63 |
| UNETR | 95.95 | 94.01 | 96.37 | **95.89** | **96.91** | 95.82 |
| VTBIS | **96.14** | **94.52** | **96.52** | 95.76 | 96.78 | **96.14** |

The visualization of the validation set prediction is illustrated in Figure 3:

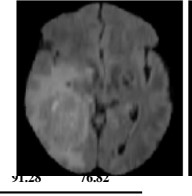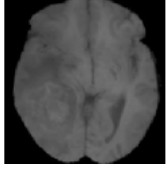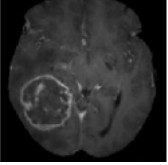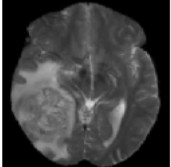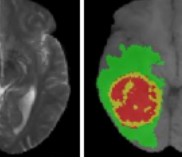

91.28    76.82

Figure 3. All the four modalities of the brain tumor visualized with the Ground-Truth and Predicted segmentation of tumor sub-regions for BraTS 2019 crossvalidation dataset. Red label: Necrosis, yellow label: Edema and Green label: Edema.

The segmentation results of our model on the Synapse multi-organ CT dataset is shown in Figure 4:

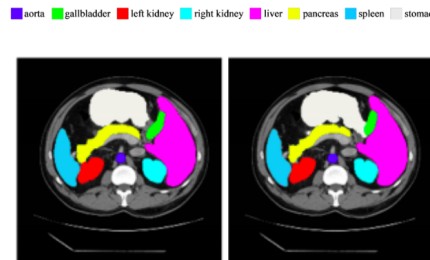

Figure 4. The segmentation results of our network on the Synapse multi-organ CT dataset. Left depicts ground truth, while the right one depicts predicted segmentation from our network.

### 4.1. Ablation Studies

The testing results of the proposed model with $224 \times 224$ and $512 \times 512$ input resolutions as input are presented in Table 6.

Table 6. Ablation study on the influence of input resolution. The best results are highlighted in bold.

| Resolution | DSC(Avg) | Aorta | Gallbladder | Kidney(L) | Kidney(R) | Liver | Pancreas | Spleen | Stomach |
|---|---|---|---|---|---|---|---|---|---|
| 224 | 78.22 | 87.53 | 63.29 | 82.53 | 78.26 | 94.53 | 55.99 | 85.38 | 76.02 |
| 512 | **84.57** | **91.00** | **67.52** | **86.18** | **83.61** | **95.84** | **70.45** | **88.68** | **83.57** |

We conduct the experiments of our model with bilinear interpolation and transposed convolution on Synapse dataset. The experiment shows that our network using transposed convolution layer achieves better segmentation accuracy.

Table 7. Ablation study on the impact of the up-sampling. Here BI denotes bilinear interpolation, TC denotes transposed convolution. The best results are highlighted in bold.

| Up-sampling | DSC | Aorta | Gallbladder | Kidney(L) | Kidney(R) | Liver | Pancreas | Spleen | Stomach |
|---|---|---|---|---|---|---|---|---|---|
| BI | 77.24 | 82.04 | 67.18 | 80.52 | 73.79 | 94.05 | 55.74 | 86.71 | 72.50 |
| TC | **78.53** | **84.55** | **68.02** | 82.46 | 74.41 | **94.59** | 55.91 | **89.25** | **73.96** |

Different skip connections values of 0, 1, 2 and 3 are used respectively. The segmentation performance of the model

increases with the increase in the number of skip connections as shown in Table 8:

Table 8. Ablation study on the impact of the number of skip connection. The best results are highlighted in bold.

| SC | DSC | Aorta | Gallbladder | Kidney(L) | Kidney(R) | Liver | Pancreas | Spleen | Stomach |
|----|-------|-------|-------------|-----------|-----------|-------|----------|--------|---------|
| 0 | 73.13 | 78.72 | 54.06 | 78.26 | 76.78 | 93.54 | 47.02 | 85.24 | 72.06 |
| 1 | 75.77 | 83.34 | 61.46 | 82.17 | 80.13 | 94.45 | 54.26 | 86.17 | 75.90 |
| 2 | 79.54 | 86.16 | 67.27 | 84.70 | 81.32 | 94.94 | 56.32 | 89.35 | 77.50 |
| 3 | **82.05** | **86.26** | **67.51** | **85.18** | **81.50** | **95.20** | **57.16** | **91.64** | **77.52** |

We explore our network at various model scales (i.e. depth (L) and embedding dimension (d)). We show ablation study to verify the impact of Transformer scale on the segmentation performance. Our network with d = 384 and L = 4 achieves the best scores of ET, WT and TC. Increasing the depth and decreasing the embedding dimension gives better results. However, the impact of depth on performance is much more than that of embedding dimension as shown in Table 9:

Table 9. Ablation study demonstrating the effect of depth and embedding dimension on our transformer. DS represents Dice score. The best results are highlighted in bold.

| Depth (L) | Embedding dim (d) | ET(DS%) | WT(DS%) | TC(DS%) |
|-----------|-------------------|---------|---------|---------|
| 1 | 384 | 69.24 | 84.16 | 70.18 |
| 1 | 512 | 69.05 | 83.87 | 69.92 |
| 2 | 384 | 70.59 | 84.88 | 72.51 |
| 2 | 512 | 70.13 | 84.15 | 71.99 |
| 4 | 384 | **72.06** | **85.39** | **73.67** |
| 4 | 512 | 71.55 | 85.06 | 73.05 |

## 5. Conclusions

Biomedical image segmentation is a challenging problem in medical imaging. Recently deep learning methods leveraging both CNN and transformer based architectures have been highly successful in this domain. In this paper, we propose a novel network named Vision Transformer (VTBIS) for Biomedical Image Segmentation. We use multi scale mechanism to split the features employing different convolutions and concatenating those individual feature maps produced before being passed to transformer blocks in encoder. The decoder also uses similar mechanism with skip connections connecting the encoder and decoder transformer blocks. The output feature map after split and concat operator is passed through a linear projection block to produce the output segmentation map. Using Dice Score and the Hausdorff Distance on multiple datasets, our network outperforms most of the previous CNN as well as transformer based architectures. In the future, we would like to use multi scale vision transformer to tackle other problems in computer vision like depth estimation.

### Acknowledgments

We would like to thank Nvidia for providing the GPUs for this work.

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
