# OpenReview forum: "VTBIS: Vision Transformer for Biomedical Image Segmentation"
_MICCAI.org/2021/Workshop/COMPAY — Reject_

### Official Review · Reviewer_gzHi · 2021-08-19
**It may not be suitable for this workshop**

**Rating:** 1
**Confidence:** 4

**Review:**

In my opinion, this submission does not fit well with the theme of the workshop. Hence, the technical contribution of this work has not been accessed. I would recommend submitting this work to other relevant venues.

It also seems this work has been submitted to multiple venues ( on the first page it says, WACV submission at the top and Anonymous ICCV submission, just below the title)

---

### Official Review · Reviewer_pAKF · 2021-08-20
**Biomedical imaging but not digital pathology paper; out of scope**

**Rating:** 3
**Confidence:** 3

**Review:**

This is a fair paper but I do not two main issues for consideration in the COMPAY workshop. First, this paper is not about digital pathology and as such not in scope. Second, the method section is very short and the methods can certainly not be reproduced from the textual description. It is not entirely clear what is new, or to what extent the proposed changes are novel. Based on these two considerations, I cannot recommend to accept this paper for the COMPAY workshop.

---

### Decision · Program_Chairs · 2021-08-25

Reject